# 7-Acetoxycoumarin Inhibits LPS-Induced Inflammatory Cytokine Synthesis by IκBα Degradation and MAPK Activation in Macrophage Cells

**DOI:** 10.3390/molecules25143124

**Published:** 2020-07-08

**Authors:** Taejin Park, Jin-Soo Park, Ji Han Sim, Seung-Young Kim

**Affiliations:** 1Department of Pharmaceutical Engineering & Biotechnology, Sunmoon University, Chungnam 31460, Korea; bark.taejin@gmail.com (T.P.); l0vely4@naver.com (J.H.S.); 2Natural Product Informatics Research Center, KIST Gangneung Institute of Natural Products, Korea 9 Institute of Science and Technology (KIST), Gangwon-do 25451, Korea; jinsoopark@kist.re.kr

**Keywords:** 7-acetoxycoumarin, umbelliferone, anti-inflammatory, acetylation, phosphorylation

## Abstract

Acetylation involves the chemical introduction of an acetyl group in place of an active hydrogen group into a compound. In this study, we synthesized 7-acetoxycoumarin (7AC) from acetylation of umbelliferone (UMB). We examined the anti-inflammatory properties of 7AC in lipopolysaccharide (LPS)-treated RAW 264.7 macrophage cells. The anti-inflammatory activity of 7AC on viability of treated cells was assessed by measuring the level of expression of NO, PGE_2_ and pro-inflammatory cytokines, namely interleukin-1β (IL-1β), interleukin-6 (IL-6) and tumor necrosis factor-α (TNF-α) in 7AC-treated RAW 264.7 macrophages. The 7AC was nontoxic to cells and inhibited the production of cytokines in a concentration-dependent manner. In addition, its treatment suppressed the production of pro-inflammatory cytokines in a dose-dependent manner and concomitantly decreased the protein and mRNA expressions of inducible NO synthase (iNOS) and cyclooxygenase-2 (COX-2). Moreover, the levels of the phosphorylation of mitogen-activated protein kinase (MAPK) family proteins such as extracellular signal-regulated kinase (ERK), c-Jun *N*-terminal kinase (JNK), p38 and nuclear factor kappa B (NF-κB) were reduced by 7AC. In conclusion, we generated an anti-inflammatory compound through acetylation and demonstrated its efficacy in cell-based in vitro assays.

## 1. Introduction

Inflammation is an early immune reaction mediated by cytokines secreted from immune cells in the host defense against viral and fungal infections and external damage, such as noxious physical or chemical stimuli. It is a highly regulated, self-limiting process for identifying and destroying invading pathogens and restoring normal tissue structure and function [1,2,3,4].

In macrophages, lipopolysaccharide (LPS)—a major form of endotoxin—induces the production of inflammatory cytokines, such as tumor necrosis factorα (TNF-α) and interleukin-1β (IL-1β) and inflammatory mediators, such as nitric oxide (NO), which is synthesized by inducible NO synthase (iNOS) and consequently cause clinical symptoms [5,6,7]. NO, a key signal molecule in many biologic processes is produced from l-arginine via the catalytic action of iNOS [8,9]. Overproduction of NO by iNOS forms reactive nitrogen species, resulting in cell death in surrounding tissues and disruption of tissue homeostasis [10,11]. Cyclooxygenase (COX) exists in the form of two isozymes, COX-1 and COX-2, and is a key enzyme in the biosynthetic pathway of PGE2 and COX-2 facilitates the conversion of arachidonic acid into prostaglandin, which in turn upregulate the expression of PGE2, another mediator of inflammation, known to induce cancer by activating angiogenesis [12,13,14].

Nuclear factor-kappa B (NF-κB), a nuclear transcription factor, is the crucial mediator in the inflammatory process and was widely implicated in inflammatory diseases, largely based on its central role in the expression of genes encoding proinflammatory cytokines. The mitogen-activated protein kinases (MAPKs), including extracellular signal-regulated kinase (ERK), c-jun *N*-terminal kinase (JNK) and p38 also mediate inflammatory and immune responses, which can be activated by LPS stimulation. Several studies have shown that MAPKs play critical roles in the activation of NF-κB [15,16,17,18,19,20]. The activation of these signaling pathways in turn activates a variety of transcription factors that control the expression of genes involved in inflammation, including iNOS and COX-2. Accordingly, MAPKs and NF-κB are important targets for anti-inflammatory molecules and many putative anti-inflammatory therapies are based on the inhibition of their activity [21,22].

Coumarin is a phenolic substance with a simple structure, widely found in vascular plants and reported to have a defensive role against insects, fungi and bacteria [23]. One of the various coumarin variants, umbelliferone (UMB) is synthesized by bacteria, fungi and shikimic pathways present in plants [24]. The Umbelliferae family includes economically important herbs such as sanicle, alexanders, angelica, asafoetida, celery, cumin, fennel, parsley and giant hogweed. The name Umbelliferone was derived from the Umbelliferae family of plants, named for their umbrella-shaped inflorescence [25,26]. The plant-derived phenolic coumarins have been indicated to play a role as dietary antioxidants because of their presence in fruits and vegetables [27]. UMB is a 7-hydroxycoumarin that is a pharmacologically active agent. By virtue of its structural simplicity, UMB was generally accepted as the parent compound for the more complex coumarins and is widely used as a synthon for a wider variety of coumarin heterocycles [28,29]. UMB has also been reported to have antifungal, anti-inflammatory, antioxidant and anticancer activities [30].

In efforts to develop anti-inflammatory drugs, UMB was converted through acetylation described earlier [31]. Acetylation is the process of chemically by substituting an acetyl group for an active hydrogen atom in a compound [32]. The resulting product now has an acetoxy group, an ester. The most commonly used acetylating agent is acetic anhydride. The aim of this study was to carry out the acetylation of UMB to 7-acetoxycoumarin (7AC) and to investigate the anti-inflammatory activity of 7AC to establish its functional properties by examining the expression levels of various inflammatory factors.

## 2. Results

### 2.1. Acetylation of UMB

In this study, a hydrogen atom of UMB was substituted by an acetyl group through acetylation. The reaction products were detected and analyzed by HPLC (Figure 1A). The acetylation reaction was examined by electrospray ionization mass spectrometry (ESI/MS) and the resulting 7AC was purified by preparative HPLC.

### 2.2. NMR Result

The ^1^H- and ^13^C-NMR analyses revealed the acetylated product as 7-acetoxycoumarin (7AC) along with the following mass data, supporting identical NMR data in previous literature [33].

7-acetoxycoumarin (7AC, 1): ^1^H-NMR (500 MHz, chloroform-d) δ 7.71 (d, *J* = 9.5 Hz, 1H), 7.50 (d, *J* = 8.4 Hz, 1H), 7.13 (d, *J* = 2.2 Hz, 1H), 7.07 (dd, *J* = 8.4, 2.2 Hz, 1H), 6.41 (d, *J* = 9.5 Hz, 1H), 2.35 (s, 3H). ^13^C-NMR (125 MHz, chloroform-d) δ 168.8, 160.4, 154.7, 153.2, 142.9, 128.6, 118.5, 116.7, 116.1, 110.5, 21.2 (Appendix A). HR-ESI/MS: *m/z* [M + H] 205.0507, calcd. 205.0495.

### 2.3. Cytotoxic Effects and NO Production of Compounds on RAW 264.7 Cells

To check the effects of 7AC, RAW 264.7 macrophage cells were first stimulated by LPS and then treated by 7AC at various concentrations. The cell viability through the MTT assay showed that 7AC was nontoxic to RAW 264.7 cells at the treated concentrations (Figure 2A). Further, 7AC treatment at the indicated concentrations reduced NO production in a dose-dependent manner (Figure 2B).

### 2.4. Production of Proinflammatory Cytokines

The effect of 7AC on the production of proinflammatory cytokines (TNF-α, IL-1β, IL-6) in RAW 264.7 cells was assessed using respective ELISA kits. On treating the cells with 7AC at 50, 100 and 200-µM concentration, IL-1β production decreased by 70%, 75% and 80%, respectively (Figure 3A). Likewise, 7AC treatment reduced IL-6 expression in a dose-dependent manner. However, 7AC only reduced TNF-α at 200 µM (Figure 3B,C). These results indicate that 7AC inhibits the production of proinflammatory cytokines, thus inhibiting the inflammatory response.

### 2.5. The Protein Expression and mRNA Levels of iNOS and COX-2

RAW 264.7 macrophage cells were treated with various concentrations of 7AC (50, 100 and 200 µM) with or without LPS (1 µg/mL) for 24 h and the results reveal that the production of PGE_2_ was inhibited by 10%, 45% and 53%, respectively (Figure 4A). Next, to determine whether the inhibitory effect of 7AC on NO and PGE_2_ production was due to the suppression of iNOS and COX-2 expression, the protein and mRNA expression of these enzymes were measured. The 7AC significantly inhibited the expression of iNOS at the treated concentrations from 50 to 200 µM (Figure 4B,D) and that of COX-2 at concentrations from 50 to 200 µM (Figure 4C,E) relative to the group treated with LPS only in a dose-dependent manner. These results, therefore, indicate that the reduction of iNOS and COX-2 is key to the decreased expression of NO and PGE_2_.

### 2.6. Expression of MAPK and NF-κB Pathways

The effects of 7AC on inflammatory signaling pathways in LPS-stimulated RAW 264.7 macrophage cells were studied by measuring the proteins of MAPK and NF-κB signaling pathways. ERK and p38 exhibited decreased phosphorylation on 7AC treatment in a dose-dependent manner (Figure 5A,B). Likewise, phosphorylation of JNK was significantly diminished by 7AC (Figure 5C). These results also demonstrated that 7AC treatment may affect the NF-κB signaling pathway by increasing the level of IκB-α expression only at 200 µM and decreasing the level of NF-κB phosphorylation (Figure 5D,E). Therefore, as per these results, 7AC inhibits the expression of proinflammatory cytokines and inflammatory mediators at least through the MAPK and NF-κB signaling pathways.

## 3. Discussion

The macrophages are immune cells that are known to play a crucial role in chronic inflammatory reactions through the creation of proinflammatory factors such as NO, PGE2, cytokines, etc. Although inflammatory factors can induce an immune response, their overexpression can damage tissue or cell [34,35].

In this study, we converted UMB to 7AC, which was characterized by MS and NMR post-acetylation [31]. Then, we determined the viability of RAW 264.7 cells for 7AC and found that 7AC is not toxic at concentrations up to 200 μM. Therefore, we investigated the anti-inflammatory effects of 7AC at the tested dosages (50, 100 and 200 μM). Using LPS-induced macrophages, we determined the effects of 7AC on the expression and secretion of NO and PGE as well as the protein and mRNA expression of the iNOS and COX-2 and observed that treatment with 7AC reduced the secretion of NO and PGE and their respective enzymes in a dose-dependent manner. These results, therefore, suggest that the inhibition of NO and PGE_2_ production was due to the decline in mRNA expression of iNOS and COX-2. Additionally, there was a decline in the levels of proinflammatory cytokines (IL-1β, IL-6 and TNF-α), which cause tissue damage and play a key role in mediating various inflammatory diseases.

The MAPK signaling pathways are pivotal signal transduction systems involved in diverse biologic responses, including the regulation of inflammatory genes through the phosphorylation of several factors associated with immune responses. To explore the possible mechanism of 7AC-induced anti-inflammatory effects, we examined the key kinases of NF-κB and MAPK pathways. The phosphorylation of IκB and p65 signifies the degradation of IκB and the activation of NF-κB. The activated NF-κB then translocates into the nucleus and increases the rate of transcription of inflammatory genes. Here, LPS-induced phosphorylation of ERK, JNK, and p38 was found to decrease on 7AC treatment, indicating that the activation of MAPK pathways was inhibited. On the exposure of macrophages to external stimuli, IκBα is released from the NF-κB complex and is degraded. Accordingly, we observed a higher level of IκB-α degradation in the LPS only-treated cells and a concentration-dependent increase in the 7AC-treated cells (Figure 5E), indicating that the presence of 7AC inhibited the phosphorylation of MAPK and the degradation of IκB-α. The result further suggests that 7AC inhibited the expression of NO, PGE_2_, iNOS, COX-2 and inflammatory cytokines, and thus, has the potential to be developed as an agent to treat and prevent inflammatory diseases or inflammatory effects.

## 4. Materials and Methods

### 4.1. Acetylation of UMB

In a typical acetylation experiment, UMB (Sigma-Aldrich, St. Louis, MA, USA) was dissolved in acetic anhydride (Samchun Chemical Co., Ltd., Seoul, Korea), and then pyridine (Junsei Chemical Co., Ltd., Tokyo, Japan) was added. The solution was kept overnight at 90 °C under magnetic stirring. After cooling slowly, the mixture was thoroughly washed with ethanol and acetone to remove the unreacted acetic anhydride and acetic acid by product. The final product was then dried at 40 °C for 1 h under vacuum.

### 4.2. HPLC Analysis and Purification of 7-Acetoxycoumarin

For HPLC analysis, a Shimadzu SpectroMonitor 3200 digital UV-Vis detector equipped with a Shim-pack GIS 0.5 mm ODS C18 column (250 × 4.6 mm I.D.) was used (Shimadzu, Kyoto, Japan). The mobile phase consisted of water with 0.1% TFA (Solvent A) and acetonitrile (Solvent B). The gradient method was used with a flow rate of 1 mL/min and the analysis was done by increasing solvent B from 10% to 100% for 30 min.

### 4.3. LCMS and NMR Analysis 7-Acetoxycoumarin

For high-resolution of the determination of the exact mass of different peaks, HR-QTOF ESI/MS was performed in positive ion mode using an ACQUITY (UPLC, Waters Corp., Milford, MA, USA) coupled with an SYNAPT G2-Si column (Waters Corp., Milford, MA, USA). The obtained mass data were subsequently analyzed by MassLynx version 4.1. NMR spectra were measured using a VNMRS 500 NMR spectrometer (Agilent Technology, Santa Clara, CA, USA) and residual solvent peaks (chloroform-*d*_6_ = δH 2.50) of deuterated NMR solvents (Sigma-Aldrich, St. Louis, MO, USA) were used as reference peaks.

### 4.4. Cell Culture and Viability Assay

RAW 264.7 macrophage cells were obtained from the Korean Cell Line Bank (KCLB, Seoul, Korea). Cells were cultured in Dulbecco’s Modified Eagle’s medium (DMEM) supplemented with 10% heat-inactivated FBS with 1% penicillin and streptomycin and placed a humidified incubator in a 5% CO2 atmosphere at 37 °C. At 80%–90% confluence, cells were plated at a density of 1.0 × 10^5^ cells/well in 24-well plates and incubated for 24 h. Cells were treated with varying concentrations of 7AC (50, 100 and 200 µM) with or without LPS (1 µg/mL) for 24 h. Cell viability was measured using the MTT assay. MTT reagent (Sigma-Aldrich, St. Louis, MO, USA) was added at a concentration of 1 mg/mL to each well, and cells were then incubated for 3 h. Subsequently, formazan crystals dissolved in DMSO were added and absorbance at 570 nm was read using a microplate reader (Spectrophotometer, Thermo Fisher, MA, USA).

### 4.5. Determination of NO and PGE_2_ Production

RAW 264.7 cells were plated at a density of 1.0 × 10^5^ cells/well in 24-well plates and incubated for 24 h. Cells were treated with varying concentrations of 7AC (50, 100 and 200 µM) with or without LPS (1 µg/mL) for 24 h. After incubation, cell culture supernatant (100 µL) was mixed with Griess reagent (100 µL; Sigma-Aldrich, St. Louis, MA, USA) and absorbance was determined at 540 nm to measure NO production. The PGE_2_ concentration in the culture supernatant of 7AC (50, 100 and 200 µM)-treated cells was estimated using a PGE_2_ ELISA Kit (Mouse PGE_2_, R&D Systems, Minneapolis, MN, USA).

### 4.6. Determination of TNF-α, IL-1β and IL-6 Production

RAW 264.7 cells were plated at a density of 1.0 × 10^5^ cells/well in 24-well plates and incubated 24 h. Cells were treated with varying concentrations of 7AC (50, 100 and 200 µM) with or without LPS (1 µg/mL) for 24 h. The concentrations of pro-inflammatory cytokines (TNF-α, IL-1β and IL-6) in culture supernatant were determined using respective ELISA kits (Mouse TNF alpha ELISA Kit, Invitrogen, Carlsbad, CA, USA; Mouse IL-6 ELISA Kit, San Diego, BD, USA; Mouse IL-1β/IL-1 F2, R&D Systems, Minneapolis, MN, USA).

### 4.7. Quantitative Reverse Transcription-Polymerase Chain Reaction (qRT-PCR) Analysis

Total RNA obtained from RAW264.7 cells was isolated using an RNA extraction kit (RNeasy Mini Kit, QIAGEN, Hilden, Germany). Total RNA (1 µg) was reverse-transcribed using a cDNA synthesis kit (PrimeScript 1st strand cDNA Synthesis Kit, TaKaRa, Kyoto, Japan). qRT-PCR was performed using the SYBR Mixture (TB Green Premix Ex Taq II, TaKaRa, Kyoto, Japan) and the primers used were as follows: iNOS-(F) AATGGCAACATCAGGTCGGCCATCACT; iNOS-(R) GCTGTGGTCACAGAAGTCTCGAACTC; COX-2-(F) GGAGAGACTATCAAGATAGT; COX-2-(R) ATGGTCAGTAGACTTTTACA; GAPDH-(R) GGTTTCTCCAGGCGGCA; GAPDH-(F) GGCATGGCCTTCCGTGT.

### 4.8. Western Blot Analysis

The total protein was extracted from the cells using RIPA buffer (Bio-rad, Hercules, California, CA, USA) and measured with a Bradford assay kit (Pierce BCA protein assay kit, Thermo Scientific, Waltham, MA, USA). Twenty micrograms of total protein samples were resolved by 10% SDS-PAGE gel and transferred onto polyvinylidene difluoride (PVDF) membranes (Bio-rad, Hercules, California, CA, USA) at 150 V and 2 h. Membranes were blocked with 5% skim milk at 2 h and incubated with primary antibody including those of COX-2 antibody (1:1000), iNOS antibody (1:5000), phospho-p44/42 MAPK (Erk1/2)(Thr202/Tyr204) antibody (1:1000), phospho-p38 MAP Kinase (Thr180/Tyr182) antibody (1:500), phospho-SAPK/JNK (Thr183/Tyr185) antibody (1:500), p44/42 MAPK (Erk1/2) antibody (1:1000), p38 MAPK antibody (1:500), SAPK/JNK antibody (1:500), phospho-NF-kB p65 (Ser536)(93H1) rabbit mAb (1:500), IκBα (L35A5) Mouse mAb amino-terminal antigen (1:500), (all the antibodies were from Cell signaling, Danvers, MA, USA) at 4 °C overnight, followed by incubation with secondary antibody (1:10,000) (HRP anti-rabbit IgG, H&L, Rockland Immunochemicals, Inc., USA) at 25 °C for 1 h. Proteins were detected using an ECL kit (Bio-rad, Hercules, California, CA, USA) and measured using image reader (LAS-4000, FUJIFILM, Tokyo, Japan).

### 4.9. Statistical Analysis

Results are expressed in terms of mean ± SD (standard deviation). The statistical significance of the differences was evaluated using the Student’s t-test for the data acquired.

## Figures and Tables

**Figure 1 molecules-25-03124-f001:**
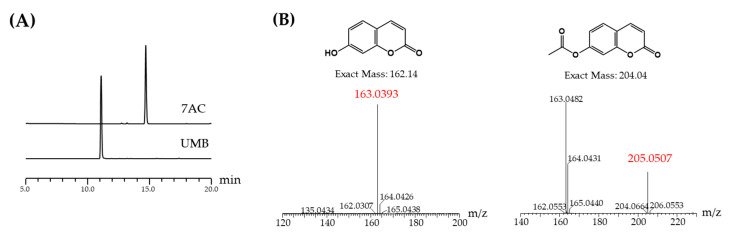
(**A**) HPLC analysis of umbelliferone (UMB), 7-acetoxycoumarin (7AC); (**B**) structure and mass analysis of UMB and 7AC.

**Figure 2 molecules-25-03124-f002:**
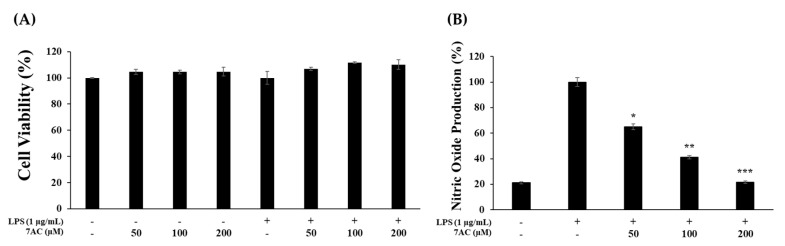
Effects of 7AC on cell viability and nitric oxide production in lipopolysaccharide (LPS)-stimulated RAW 264.7 cells. (**A**) Cell viability was assessed in cells that were not stimulated or stimulated with LPS (1 μg/mL) in the present of 7AC for 24 h; (**B**) nitric oxide (NO) production was determined using the Griess reagent method. The data represent the mean ± SD of triplicate experiments. * *p* < 0.05, ** *p* < 0.01, *** *p* < 0.005 versus LPS alone.

**Figure 3 molecules-25-03124-f003:**
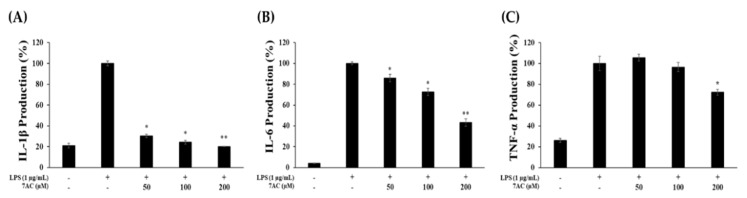
Effect of 7AC on (**A**) TNF-α, (**B**) IL-1β and (**C**) IL-6 production in LPS -stimulated RAW 264.7 cells. Cells were stimulated with 1 µg/mL of LPS only or with LPS along with varying concentrations (50, 100 and 200 µM) of 7AC for 24 h. Their production was determined by ELISA. The data represent the mean ± SD of triplicate experiments. * *p* < 0.05, ** *p* < 0.01 versus LPS alone.

**Figure 4 molecules-25-03124-f004:**
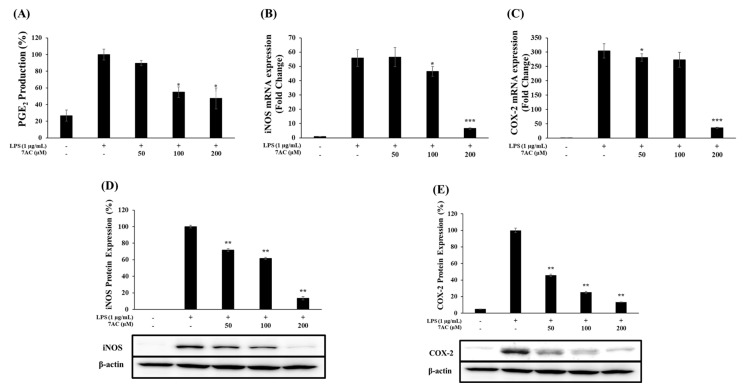
Effects of 7AC on PGE_2_ production and the protein and mRNA levels of iNOS and COX-2 in LPS-stimulated RAW 264.7 cells. (**A**) Production of PGE_2_ was assayed in the culture medium of cells stimulated with LPS (1 μg/mL) for 24 h in the presence of 7AC (50, 100 and 200 µM) by ELISA; (**B**,**C**) mRNA and (**D**,**E**) protein levels of iNOS and COX-2 were determined by qRT-PCR and western blot, respectively. The data represent the mean ± SD of triplicate experiments. * *p* < 0.05, ** *p* < 0.01, *** *p* < 0.001 versus LPS alone.

**Figure 5 molecules-25-03124-f005:**
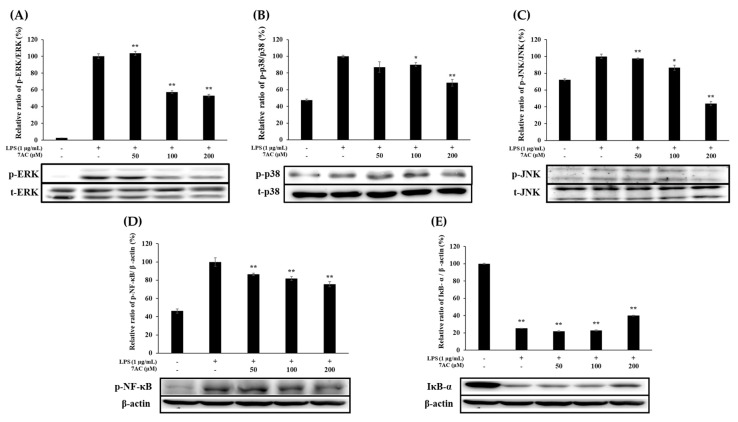
Effect of 7AC on the protein levels of (**A**) ERK, (**B**) p38, (**C**) JNK, (**D**) NF-κB and (**E**) IκB kinase in LPS-stimulated RAW 264.7 cells. Cells (5.0 × 10^5^ cell/mL) were stimulated with LPS (1 µg/mL) in the presence of 7AC (50, 100 and 200 µM) for 40 min. Whole-cell lysates (20 µg) were prepared and the protein level was subjected to 10% SDS-PAGE and expressions of MAPK, NF-κB and β-actin were determined by western blotting. * *p* < 0.05, ** *p* < 0.01 versus LPS alone.

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
