# Peer review of "7-Acetoxycoumarin Inhibits LPS-Induced Inflammatory Cytokine Synthesis by IκBα Degradation and MAPK Activation in Macrophage Cells"

_molecules, 2020, doi:10.3390/molecules25143124_

Round 1

Reviewer 1 Report

The manuscript by Taejin Park et al. describes the antiiflammatory effects of 7-Acetoxycoumarin through different biological assays.

The authors do not mention why they have choosen this compound.

In order to have a complete study other derivatives of umbelliferone or coumarin should be studied in the same assays. The anti-inflammatory effect of umbelliferone is well known.

Additionally the spectroscopic characterization of this derivative is missing (13C-NMR).

Metabolism or toxicological studies have not been conducted.

The manuscript presents problems in language and style.

In general the presented results are very limited thus I do not recommend its publication in this form.

Author Response

We fully agree with what you have pointed out. And also we tried to do our best to give an answer of reviewer’s pointed out. All of our authors are unanimously honored that International Journal of Molecular Science has given us a chance to evaluate our research in a world-wide journal. I know it is our greed, but if you point out additional points, I will be able to modify our paper in good faith. Thank you.

Point 1: The authors do not mention why they have chosen this compound.

Response 1: Although 7-Acetoxycoumarin (7AC), a coumarin derivative, functions as protease inhibitors and HIV inhibitors (Patent US6005103), its anti-inflammatory properties were less investigated.

Point 2: In order to have a complete study other derivatives of umbelliferone or coumarin should be studied in the same assays. The anti-inflammatory effect of umbelliferone is well known.

Response 2: The cell viability of coumarin (10, 50, and 100 µM) in 1 µg/mL LPS-treated RAW264.7 cells was decreased to 104, 97, 92% in a concentration-dependent manner [Ref. 1] while that of 7AC at 50, 100, and 200 µM remains > 100% [Figure 2 of the manuscript]. Both coumarin and 7AC, when each treated at 50 µM, inhibited LPS-induced NO production by about 40%. However, 50 µM of 7AC inhibited LPS-induced IL-1β production by about 70% while 50µM of coumarin inhibited IL-1β by about 6% [Figure 3 of the manuscript and Ref. 1]. Therefore, compared to coumarin, 7AC is more efficacious in the inhibition of LPS-induced IL-1β and showed lesser cytotoxicity.

Reference

[1] Sandhiutami, N.M.D.; Moordiani, M.; Laksmitawati, D.R.; Fauziah, N.; Maesaroh, M.; Widowati, W. In vitro assessment of anti-inflammatory activities of coumarin and Indonesian cassia extract in RAW264.7 murine macrophage cell line. Iran J Basic Med Scim. 2017, 20, 99-106.

[2] Moossavi, M.; Parsamanesh, N.; Bahrami, A.; Atkin, S.L.; Sahebkar, A. Role of the NLRP3 inflammasome in cancer. Molecular Cancer. 2018, 17, 158.

Point 3: Additionally the spectroscopic characterization of this derivative is missing (13C-NMR).

Response 3: We completely revised them. Please see “line 86-91”. The added part was marked in red.

Point 4: Metabolism or toxicological studies have not been conducted.

Response 4: We are totally dependent on your generous mind. But, it is an experiment that cannot be done in our laboratory at present. In the next experiment, we will reflect the feedback of reviewer. We look forward to your generosity.

Point 5: The manuscript presents problems in language and style.

Response 5: We have revised the WHOLE manuscript carefully and tried to avoid any grammar or syntax error. In addition, we brought a submitted paper to a native speaker of English (Editage; http://www.editage.com), Job code: KRHDH_7. We believe that the language is now acceptable for the review process.

Reviewer 2 Report

The Manuscript entitle “7-Acetoxycoumarin Inhibits LPS-Induced Inflammatory Cytokine Synthesis by IκBα Degradation and MAPK Activation in Macrophage Cells” is well written, with the experiments supporting the results taken. This study refers the effect of 7-Acetoxycoumarin (7AC) as potential anti-inflammatory agent, which was unknown until the date. However, the authors found that the majority of the effect of 7AC happened at higher concentrations, at 100μM or 200μM. Please justify why it was selected those concentrations, since these concentrations are relative high for 7AC to possible become a clinic agent to treat and prevent inflammatory effects.

In addition, some other points should be addressed:

  • In Fig 2A, it is missing the conditions regarding the toxicity of the different concentrations of 7AC at their own (meaning toxicity of 7AC without the presence of LPS induction). The authors mentioned that “7AC was non-toxic to RAW 264.7 cells at the treated concentrations”, thus this has to be demonstrated.
  • In line 103, the author state “…7AC treatment reduced IL-6 and TNF-a expression in a dose-dependent manner”. However, 7AC only reduced TNF-a at 200μM. Please correct.
  • In line 119, please correct the concentrations. Instead of being, “12.5 to 100μM” should be “50 to 200μM”.
  • In line 139, Change “Figure 2” for Figure 5”.
  • All the graphs corresponding to the protein expression quantification have the statistical analysis missing, named in Figures 4D, 4E, 5A, 5B, 5C, 5D, 5E. Please add statistic to those graphs.
  • In Figure 5A, 5B and 5C, the quantification of the proteins should be expressed in protein phosphorylation per total protein (and not just the quantification of the phospho protein). Please confirm if this analysis was done, since it is not clear in the YY axis.
  • In the lines 134-135, please reformulate the sentence, since 7AC increased the level of IҡB-β expression only at 200μM. The same it seems for NF-ҡB phosphorylation, which 7AC seems to decrease only at 200μM (it requires statistic).
  • In line 134, “These results thus suggest that” has no sense. Change for example for “The results also demonstrated that”.
  • In lines 136-137 state “Therefore, as per these results, 7AC inhibits the expression of pro-inflammatory cytokines and inflammatory mediators through the MAPK and NF-κB signaling pathways”. Please add “at least” meaning “….inflammatory mediators at least through the MAPK…”, since the authors did not analyze all the signaling pathways involved.

Author Response

We tried to do our best to give an answer of reviewer’s opinions. In addition we are unanimously honored that International Journal of Molecular Science has given us a chance to evaluate our research in a world-wide journal. Thank you for your consideration and kindness.

Point 1: In Fig 2A, it is missing the conditions regarding the toxicity of the different concentrations of 7AC at their own (meaning toxicity of 7AC without the presence of LPS induction). The authors mentioned that “7AC was non-toxic to RAW 264.7 cells at the treated concentrations”, thus this has to be demonstrated.

Response 1:  The cell viability of 7AC resulted in non-toxic results at 50, 100 and 200 µM [added Figure 1A of the manuscript]. But, Effects of LPS on cell viability [ref. 1]. We first investigated the effect of LPS (1, 2, 4 and 8 µg/ml) on the cell proliferation of RAW 264.7 macrophages using MTT assay. LPS suppressed the proliferation of RAW 264.7 macrophages at 4-8 µg/ml in a dose-dependent manner. So, LPS at 1 μg/ml, a nontoxic concentration, was used in experiments. In the lines 99-100, the revised part was marked in red.

Reference

[1] Liu, Y.; Su, W.W.; Wang, S.; Li, P.B. Naringin inhibits chemokine production in an LPS-induced RAW 264.7 macrophage cell line. Mol. Med. Rep. 2012, 6: 1343–1350.

Point 2: In line 103, the author state “…7AC treatment reduced IL-6 and TNF-a expression in a dose-dependent manner”. However, 7AC only reduced TNF-a at 200μM. Please correct.

Response 2: In the lines 107, as reviewer’s pointed out, we completely fixed it.

Point 3: In line 119, please correct the concentrations. Instead of being, “12.5 to 100μM” should be “50 to 200μM”.

Response 3: In the lines 123, as reviewer’s pointed out, we completely fixed it.

Point 4: In line 139, Change “Figure 2” for Figure 5”.

Response 4: In the lines 145, as reviewer’s pointed out, we completely fixed it.

Point 5: All the graphs corresponding to the protein expression quantification have the statistical analysis missing, named in Figures 4D, 4E, 5A, 5B, 5C, 5D, 5E. Please add statistic to those graphs.

Response 5: I revised it according to your opinion.

Point 6: In Figure 5A, 5B and 5C, the quantification of the proteins should be expressed in protein phosphorylation per total protein (and not just the quantification of the phospho protein). Please confirm if this analysis was done, since it is not clear in the YY axis.

Response 6: I revised it according to your opinion.

Point 7: In the lines 134-135, please reformulate the sentence, since 7AC increased the level of IκB- α expression only at 200μM. The same it seems for NF-κB phosphorylation, which 7AC seems to decrease only at 200μM (it requires statistic).

Response 7: In the lines 139, as reviewer’s pointed out, we completely revised it.

Point 8: In line 134, “These results thus suggest that” has no sense. Change for example for “The results also demonstrated that”.

Response 8: In the lines 138, as reviewer’s pointed out, we completely revised it. Thank you.

Point 9: In lines 136-137 state “Therefore, as per these results, 7AC inhibits the expression of pro-inflammatory cytokines and inflammatory mediators through the MAPK and NF-κB signaling pathways”. Please add “at least” meaning “….inflammatory mediators at least through the MAPK…”, since the authors did not analyze all the signaling pathways involved.

Response 9: In the lines 141, as reviewer’s pointed out, we have revised it.

Reviewer 3 Report

Dear, Seung-Young Kim, Ph.D.

Please check and revise.

-------------------------------------------------------------------------------------------------------------------------

7AC: It’s a known compound. Which ref. are you comparing with NMR data?

Fig 1 (B): Exact mass, it’s necessary four digit.

Have you been trying only 7AC? I recognize trying some acylated analogus.

-------------------------------------------------------------------------------------------------------------------------

Author Response

We tried to do our best to give an answer of reviewer’s opinions. In addition we are unanimously honored that International Journal of Molecular Science has given us a chance to evaluate our research in a world-wide journal. Thank you for your consideration and kindness.

Point 1: 7AC: It’s a known compound. Which ref. are you comparing with NMR data?

Response 1: We completely added them. Please see “line 86-91”. The added part was marked

Point 2: Fig 1 (B): Exact mass, It’s necessary four digit.

Response 2: In the lines 91, as reviewer’s pointed out, we completely revised it. Thank you.

Point 3: Have you been trying only 7AC? I recognize trying some acylated analogus.

Response 3:  We converted UMB to 7AC using pyridine and acetic anhydride through acetylation as stated in the paper. In this study, other analogous wasn't used. I'm going to trying it in the future. Thank you.

Round 2

Reviewer 1 Report

The manuscript by Taejin Park et al. describes the antiiflammatory effects of 7-Acetoxycoumarin through different biological assays.

It is a biological study not complete due to the fact that it does not contain any other derivatives evaluated in the same assays. The anti-inflammatory effect of umbelliferone is well known. Metabolism or toxicological studies have not been conducted at the revised form.

In my opinion the presented results are rather poor and I prefer to leave the decision to the Editor’s choise.

Reviewer 2 Report

Thanks for answering the questions. Nevertheless, some corrections/changes still need to be done in v2, as following:

  • Line 104: Adding the sentence in red, the following sentence “Cell viability…..24 h” (lines 104-105) is not required.
  • Line 144: remove “thus suggest” since is not required
  • Figure 4: The graphs D and E, previously had 7AC in the xx axis, and now it was changed for “sample”. Why is that?
  • Line 151: The name of the Figure 5 is still incorrect, still says “Figure 2”. Please correct for Figure 5, as already mentioned. In addition, in the legend of the Figure 5 it is missing information regarding the statistic that was added to the Figure. Also, in Graph 5E, the second bar corresponding to LPS (+) 7AC (-) has statistic, which was not supposed, since all the conditions are being compared with this condition (LPS alone).

Reviewer 3 Report

Dear, Seung-Young Kim, Ph.D.

-------------------------------------------------------------------------------------------------------------------------

You have revised.

-------------------------------------------------------------------------------------------------------------------------